# Substrate Matters: Ionic Silver Alters Lettuce Growth, Nutrient Uptake, and Root Microbiome in a Hydroponics System

**DOI:** 10.3390/microorganisms12030515

**Published:** 2024-03-04

**Authors:** LaShelle Spencer, Blake Costine, Tesia Irwin, Anirudha Dixit, Cory Spern, Angie Diaz, Brittney Lozzi, Wenyan Li, Christina Khodadad, Trent Smith, Raymond Wheeler, Aubrie O’Rourke

**Affiliations:** 1Noetic Strategies, Inc., LASSO II, Kennedy Space Center, FL 32899, USA; lashelle.e.spencer@nasa.gov (L.S.); cory.j.spern@nasa.gov (C.S.); christina.l.khodadad-1@nasa.gov (C.K.); 2Aetos Systems Inc., LASSO II, Kennedy Space Center, FL 32899, USA; blake.m.costine@nasa.gov (B.C.); anirudha.dixit@nasa.gov (A.D.); 3Astrion, LASSO II, Kennedy Space Center, FL 32899, USA; tesia.d.irwin@nasa.gov (T.I.); wenyan.li-1@nasa.gov (W.L.); 4Bennett Aerospace, LASSO II, Kennedy Space Center, FL 32899, USA; angie.m.diaz@nasa.gov; 5NASA Kennedy Space Center Office of STEM Engagement (OSTEM) Intern Program, Kennedy Space Center, FL 32899, USA; brittney.lozzi@bcm.edu; 6Program in Genetics and Genomics, Baylor College of Medicine, Houston, TX 77030, USA; 7NASA Exploration Research and Technology, Kennedy Space Center, FL 32899, USA; trent.m.smith@nasa.gov (T.S.); raymond.m.wheeler@nasa.gov (R.W.)

**Keywords:** lettuce, silver, hydroponics, arcillite, substrate, metagenomics, essential elements, space crop

## Abstract

Ionic silver (Ag^+^) is being investigated as a residual biocide for use in NASA spacecraft potable water systems on future crewed missions. This water will be used to irrigate future spaceflight crop production systems. We have evaluated the impact of three concentrations (31 ppb, 125 ppb, and 500 ppb) of ionic silver biocide solutions on lettuce in an arcillite (calcinated clay particle substrate) and hydroponic (substrate-less) growth setup after 28 days. Lettuce plant growth was reduced in the hydroponic samples treated with 31 ppb silver and severely stunted for samples treated at 125 ppb and 500 ppb silver. No growth defects were observed in arcillite-grown lettuce. Silver was detectable in the hydroponic-grown lettuce leaves at each concentration but was not detected in the arcillite-grown lettuce leaves. Specifically, when 125 ppb silver water was applied to a hydroponics tray, Ag^+^ was detected at an average amount of 7 μg/g (dry weight) in lettuce leaves. The increase in Ag^+^ corresponded with a decrease in several essential elements in the lettuce tissue (Ca, K, P, S). In the arcillite growth setup, silver did not impact the plant root zone microbiome in terms of alpha diversity and relative abundance between treatments and control. However, with increasing silver concentration, the alpha diversity increased in lettuce root samples and in the water from the hydroponics tray samples. The genera in the hydroponic root and water samples were similar across the silver concentrations but displayed different relative abundances. This suggests that ionic silver was acting as a selective pressure for the microbes that colonize the hydroponic water. The surviving microbes likely utilized exudates from the stunted plant roots as a carbon source. Analysis of the root-associated microbiomes in response to silver showed enrichment of metagenomic pathways associated with alternate carbon source utilization, fatty-acid synthesis, and the ppGpp (guanosine 3′-diphosphate 5′-diphosphate) stringent response global regulatory system that operates under conditions of environmental stress. Nutrient solutions containing Ag^+^ in concentrations greater than 31 ppb in hydroponic systems lacking cation-exchange capacity can severely impact crop production due to stunting of plant growth.

## 1. Introduction

Ionic silver (Ag^+^) is being investigated as a residual biocide for use in NASA spacecraft potable water systems on future crewed missions and has been used for many years on the Russian Mir and Russian modules of the International Space Station (ISS) [1,2]. NASA currently uses iodine (I_2_) as a biocide in the potable water system on the ISS. Iodine is removed before use because excessive amounts can lead to thyroid dysfunction [3]. The advantage of ionic silver is that it does not need to be removed from potable water before consumption. The design specifications for a silver dosing device in a spacecraft potable water system include the ability to add between 200 to 500 ppb (0.2–0.5 mg/L) of silver [2]. The final amount of silver that will be dispensed from the potable water dispenser for end use is currently unknown and may vary due to the propensity for silver to plate out of solutions onto system materials [3].

In addition to providing clean water to the crew and other life-support system functions, potable water will be used to irrigate future spaceflight crop production systems. Substrate-based watering systems are currently used on the ISS (e.g., Veggie and Advanced Plant Habitat (APH)) and use arcillite—a calcinated clay with high cation ion exchange capacity [4,5]. Surface-tension effects exhibited in reduced gravity can lead to thick boundary layers of water around plant roots and, in turn, lead to root hypoxia [6]. Substrate-free nutrient delivery systems such as hydroponics can reduce waste and up-mass in future missions [7] and can provide efficient absorption of nutrients as roots are directly in contact with the nutrient solution [8]. Recent work to evaluate this efficiency and overcome the challenges of watering plants in the microgravity environment has included spaceflight testing of hydroponics systems such as Plant Water Management (PWM) [9] and the eXposed Root On-Orbit Test System (XROOTS) [10].

In this study, the Ag^+^ in our irrigation water is leached from a foam containing silver chloride currently being tested for use in spacecraft water processor assemblies [2]. However, research concerning the effect of silver on plants has focused on silver nanoparticles (AgNPs) due to their ubiquitous use in nanomaterials. In agriculture, AgNPs have been used to stimulate plant growth, inhibit fungal growth, and modify ripening in fruits [11,12]. Conversely, AgNP exposure is known to affect plant morphology by inhibiting seed germination and root growth, while reducing biomass and leaf area and increasing the production of reactive oxygen species (ROS) in plant tissue [11]. As an example, 5–10 nm particles tested at a concentration of 100–500 ppb caused a reduction in the elongation and fresh weights of shoots in roots, total chlorophyll, protein, and sugar content of *Lupinus termis* [13]. Additionally, the uptake of AgNP in radish was associated with a decrease in the water content of the plant tissue as well as nutrients in the form of Ca, Mg, B, Cu, Mn, and Zn [14]. AgNPs also affect plant hormones and are known to interfere with root gravitropism in *Arabidopsis* by reducing the auxin receptor-related genes for the signaling hormone, auxin [15]. Conversely, cytokinin [16] and ethylene [17] levels are known to increase in response to AgNP exposure, although Ag^+^ has also long been known to block the action of ethylene in plants [18].

The mechanism of toxicity for AgNPs is attributed to the induction of oxidative stress [19,20] in plant cells which can damage cell membranes, structure, protein, and DNA through the excessive production of reactive oxygen species (ROS) from the chloroplasts, mitochondria, and peroxisomes. Water from the environment oxidizes AgNPs and Ag^+^ is released, resulting in more ROS. Characterizing the direct effect of ionic silver is important as Ag^+^ is known to be persistent, accumulate, and can become highly toxic to organisms, including plants [11]. Transcriptomic analysis has illustrated that Ag^+^ inflicts greater observable damage to membranes and DNA than intact AgNPs [21]. Overall, as the size of the AgNP becomes closer to ionic silver, the negative effects increase [11]. Furthermore, Ag^+^ is known to compete with copper (Cu^+^) for the binding of plastocyanin from the photosynthetic electron transport chain and can either reduce or inactivate the important energy-producing process [22].

The use of Ag^+^ may adversely affect plant growth in spaceflight but the choice of a substrate-based or substrate-free growth system and silver concentration could mitigate the more extreme effects. One reason for this is due to the high cation-exchange capacity of arcillite that allows for the adsorption of positively charged ions due to negatively charged surfaces. We expected that arcillite as a substrate with high cation-exchange capacity would act like biochar, which holds a lower cation-exchange capacity, in its ability to diminish the uptake of toxic metals in plants by immobilizing the metal across the surface of the substrate via ion exchange [23,24]. To that end, it remained to be determined what dosage of Ag^+^ is acceptable for plant irrigation water under ISS-like environmental conditions (~3000 ppm carbon dioxide) and whether Ag^+^ will need to be removed from potable water before use as space crop irrigation water. To answer this, we asked at what concentration does silver-treated water influence the growth, nutrient content, and microbiome of a space crop grown on an arcillite (calcinated clay) substrate or without a substrate (hydroponically)?

## 2. Materials and Methods

In this study, we investigated the effect of Ag^+^ on lettuce plant growth in ISS-relevant growth chamber conditions in both the traditional ISS growth medium of arcillite and in a hydroponics setup. The methods of this study were conducted to address the following objective: to determine at what concentration silver-treated water influences the growth, nutrient content, and microbiome of a space crop grown on an arcillite (calcinated clay) substrate or without a substrate (hydroponically).

### 2.1. Ionic Silver (Ag^+^) Irrigation Water Preparation

The design specifications for a silver dosing device in a spacecraft potable water system include the ability to add between 200 to 500 ppb (0.2–0.5 mg/L) of silver. Accordingly, we selected a dilution series of ionic silver to encompass: (1) the highest possible dose (500 ppb) to dispense from the potable water system; (2) a dose (~125 ppb) close to the acceptable drinking water dose limits of silver, which is 100 ppb [25]; and (3) the lowest dose (31 ppb) to represent a dose thought to have next to no adverse effects in a substrate-based system as reported in the literature [12]. An AgFoam [2] cartridge was used as the Ag^+^ doser and consists of a 290 mL in-line flow-through polypropylene cartridge filled with an AgFoam. The AgFoam is a polyurethane and silver chloride nanoparticle (AgClNP) composite material that releases Ag^+^ into the water via the dissolution of the AgCl. This was used to produce the three separate Ag^+^ concentrations (31 ppb, 125 ppb, 500 ppb). Deionized water was pumped through the AgFoam cartridge at a constant flow rate and the effluent was collected in 20 L carboys. The flow rate was adjusted to provide enough contact time to reach a 500 ppb concentration of Ag^+^ in the effluent water. The Ag^+^ concentration was determined using a silver ion selective electrode (ISE). From this stock silver water, the appropriate dilutions were made using deionized water and confirmed using the ISE. The final silver solutions were then filtered using a 0.22 μm filter and stored in a sterile carboy with a spigot for dispensing. The starting concentrations were measured using ISE and checked weekly to ensure no silver depletion.

### 2.2. Seed Sanitization and Storage

Seeds were sanitized via chlorine gas using the approach described in the references [4,26]. After sanitization, seeds were stored in a refrigerator at 4 °C until planting.

### 2.3. Plant Growth Environmental Conditions

Lettuce (*Lactuca sativa* cv. ‘Outredgeous’) was grown in environmental growth chambers (Percival PGW-48, Perry, IA, USA) located in the Space Station Processing Facility at NASA Kennedy Space Center, Florida (located at sea level with an atmospheric pressure of ~101 kPa). The chamber air temperature, relative humidity (RH), and CO_2_ levels were maintained at 23 °C, 50%, and 3000 ppm, respectively, for all tests. These set points were selected to simulate an average environment of the ISS. Lighting was provided by overhead light-emitting diode (LED) grow lights (Percival, Perry, IA, USA). The photosynthetic photon flux density (PPFD) target was 300 μmol m^−2^ s^−1^ averaged over the entire canopy, with a 16 h (16 h light/8 h dark) photoperiod. Spectral quality consisted of 23% blue (400–500), 27% green (500–600 nm), 50% red (600–700 nm), and 25 µmol m^−2^ s^−1^ of far red using a combination of 450 nm, 660 nm, 730 nm, and white (6000 K) LEDs.

### 2.4. Experimental Design

Treatments included one control and three dilutions of Ag^+^ residual biocide water (0 ppb/control, 31 ppb, 125 ppb, and 500 ppb). Each treatment was applied to two arcillite trays or Nutrient Film Technique (NFT) hydroponic systems (each containing 8 plants) providing *n* = 16 technical replicates. Trays/systems were randomly positioned to minimize chamber effects (Figure 1). Each experiment was repeated with three grow outs for the arcillite and NFT hydroponics systems to obtain a biological replication of three. This amounted to 16 subsample plants per control and treatment for a total of 48 samples per control and each of the treatments. Plants were grown for 28 days to achieve a mature specimen, similar to what would be ingested by the crew in an ISS operation setting.

### 2.5. Arcillite Testing

Sanitized lettuce seeds were sown in square plastic pots (9 cm tall, 10.2 cm width) containing autoclaved arcillite (Turface Athletics Pro League Elite, PROFILE Products LLC; Buffalo Grove, IL, USA) amended with 7.5 g/L T70 16-6-8 Nutricote controlled-release fertilizer (Florikan, Bowling Green, FL, USA). Arcillite consists of particles of calcined clay and has commonly been used for spaceflight testing with plants [4]. In each pot, four seeds were sown on the surface of moistened media. Then, pots were placed in trays (size: 1020; dimensions: 10.94″ W × 21.44″ L × 2.44″ D). Plots were manually irrigated and checked daily. Each tray was irrigated with the above-outlined dilutions of filter-sterilized residual biocide-treated water (Figure 1). Trays were covered with a clear dome for 2–3 days to promote germination. Once true leaves emerged, seedlings were thinned to one per pot. Arcillite growth experiments were conducted in three grow-outs to obtain a biological replication of three with 16 subsamples per control and treatment.

### 2.6. Hydroponic Testing

Sanitized lettuce seeds were sown in rockwool cubes (Grodan, Roermond, The Netherlands). Two seeds were sown per cube and then placed in Crop King desktop NFT systems (Cropking, Lodi, OH, USA). Each system was supplied with non-filter sterilized modified 0.5 × Hoagland’s/Arnon solution made with the above-outlined dilutions of filter sterilized Ag^+^ residual biocide treated water [27]. Each system had pH and electrical conductivity (EC) checked and adjusted manually daily. Additions of 1 M nitric acid were used to maintain pH levels to 5.7–5.8 and EC to 1.2 mS cm^−1^. Nutrient solution changeouts were completed weekly with freshly prepared solutions to maintain biocide treatment conditions and replenish nutrients. NFT systems were covered with a clear dome for 2–3 days to promote germination. Once true leaves emerged, seedlings were thinned to one per plug. Hydroponic growth experiments were conducted in three grow-outs to obtain a biological replication of three with 16 technical replicates per control and treatment.

### 2.7. Plant Growth Measurements

The general appearance of the plants was monitored qualitatively throughout the entire experiment and recorded in a logbook daily. At 28 days after planting (DAP), plants were harvested, and plant growth measurements were recorded. Measurements included shoot fresh mass, shoot height and diameter, leaf area, relative chlorophyll, and anthocyanin content index. Leaf area measurements were made using a Li-Cor area meter (LI-3100C, Lincoln, NE, USA), relative chlorophyll estimates were made using a SPAD-502DL meter (Konica Minolta Sensing, Osaka, Japan), and anthocyanin content index was measured using an ACM-200 Plus meter (Opti-Sciences, Inc., Hudson, NH, USA).

### 2.8. Nutrient and Vitamin Analysis of Lettuce Samples

#### 2.8.1. Elemental Nutrient Analysis

A plant specimen from each condition was randomly chosen for sample preparation from each of the three trials of the arcillite or hydroponic testing. This provided a biological replication of three for each silver treatment condition to undergo elemental nutrient analysis by ICP-OES. Dried plant samples were ground to a fine powder either by hand with a mortar and pestle or with a One-Touch Grinder (Krups, PN F20342). Dried and ground samples (0.20–0.26 g) were weighed in 70 mL HD polyethylene vials (CPI International, Santa Rosa, CA, USA). The vials were then placed on a graphite heating block open vessel system (ModBlock, PM 4370-010007, CPI International, Santa Rosa, CA, USA) with cover glasses at 95 °C. Then, an aliquot of 5 mL, 70% HNO_3_ (trace metal grade, Fisher Scientific, Suwanee, GA, USA) was added to the samples and allowed to boil for about 2 h. After cooling, 2.5 mL of 30% H_2_O_2_ (Fisher Scientific, Suwanee, GA, USA) was added and allowed to sit until the hydrogen peroxide reaction was completed. Samples were then reheated for 50 min and allowed to cool overnight before dilution with 18 Mohm water up to 50 mL. To ensure no particles of the sample remained, digested solutions were run through a 0.2 μm syringe filter prior to running on the ICP-OES instrument (Thermo Fisher Scientific iCAP 7000, Waltham, MA, USA). A multielement standard (Standard 27 (27 elements), High Purity Standards, Charleston, SC, USA) was used to establish a multielement calibration curve for the elements (SPEX standards, Metuchen, NJ, USA). Work was conducted following EPA method 200.7, revision 4.4 (1994).

#### 2.8.2. Vitamin Analysis

Vitamin B1—thiamine HCl (AOAC 942.23 mod., Most Matrices), vitamin C—ascorbic acid (AOAC 967.22 mod.), and vitamin K1 (phylloquinone) contents were analyzed by Eurofins Scientific Inc. (Eurofins, Inc., Des Moines, IA, USA) using HPLC.

### 2.9. Shotgun Metagenomics Sample Processing

Lettuce plants grown in arcillite irrigated with silver ion solution or in a hydroponic system in direct contact with silver ion solution were harvested as shoot and root at 28 DAP. A total of 108 50 mL samples of water from the hydroponic trays were taken at 0, 7, 14, 21, and 28 DAP for treatments and controls. Twelve water control samples were processed at the start of the hydroponics and arcillite experiment. Tissue and water were flash-frozen in 50 mL conical tubes in liquid nitrogen and stored at −80 °C until further use. For DNA extractions, shoot and root tissue (48 arcillite shoots, 48 arcillite root samples, 48 hydroponic shoots, and 48 hydroponic roots) were ground to a coarse powder in liquid nitrogen using a mortar and pestle. Approximately 100 mg of shoot/root tissue was processed for DNA extraction using the ZymoBIOMICS DNA Miniprep kit (Zymo Research, Irvine, CA, USA) following the manufacturer’s recommendations. A total of 108 water samples from the hydroponic trays were concentrated using the CP SELECT (0.20 µm) instrument and eluted in 750 µL 0.075% Tween 20/Tris buffer (InnovaPrep, Drexel, MO, USA). Following DNA isolation, 1 µg of each sample was processed using NEBNext^®^ Microbiome DNA Enrichment Kit (New England Biolabs, Ipswich, MA, USA) following the manufacturer’s recommendations. Microbial enriched fractions were quantified using a Qubit dsDNA HS kit (Invitrogen, Carlsbad, CA, USA). Shotgun metagenomic sequencing libraries were prepared using an Illumina DNA Prep library preparation kit with unique dual indexes via IDT for Illumina DNA/RNA UD Indexes (Illumina Inc., San Diego, CA, USA). Libraries were sequenced as paired-end 150 bp sequences on a NextSeq 1000 instrument (V1.5.0.42699) at Kennedy Space Center.

### 2.10. Shotgun Metagenomics Sequencing Data Analysis

Three hundred and twelve forward and reverse FASTQ files were analyzed for sequencing quality using FastQC ver. 0.12.1 and the reports were compiled using MultiQC ver. 1.17 [28]. Six hundred read files were processed for the removal of host reads using HoCoRT ver. 1.2.2 [29]. The 315 microbial read files remaining after this subtraction were classified using the NASA GeneLab metagenomics pipeline (https://github.com/nasa/GeneLab_Data_Processing/tree/master/Metagenomics/Illumina, accessed on 11 September 2023), which includes raw data quality control, quality filtering, and trimming and quality control followed by taxonomic and functional profiling using HUManN3 ver. 3.6 and MetaPhlAn3 ver. 4.0.1 [30]. MetaPhlAn3 was used to provide the relative abundances of microbial genera within the samples by mapping the metagenomic forward and reverse reads to a set of clade-specific marker sequences representing bacterial and archaeal phylogenies. The MetaPhlAn3 output also provided a breakdown of the genomic pathways associated with 126 metagenomic samples with associated pathway abundances. Abundance and diversity metrics were summarized using ‘phyloseq’ ver. 1.38.0 [31] in Rstudio for the same 126 samples. Relative abundance was visualized using ‘phyloseq’. Shannon index alpha diversity was output from ‘phyloseq’ and visualized using ‘ggboxplot’ of the ‘ggpubr’ ver. 0.6.0 package. DEseq2 ver. 1.34.0 [32] was used to establish the pathways that were enriched in the roots for each silver treatment according to the growth setup as compared to the 0 ppb silver root control for the respective growth setup. A log_2_FC of 2.5 and a *p*-value cutoff of 0.01 were used to establish significantly differentially expressed genetic pathways in the silver-treated metagenomes as compared to the control sample metagenomes.

## 3. Results

### 3.1. Lettuce Growth Metrics

Arcillite-grown plants were not significantly affected by watering with any of the concentrations of silver-treated water as assessed by the growth metrics of plant height, plant area, shoot fresh mass, leaf area, leaf number shoot dry mass, relative chlorophyll content, or anthocyanin content (Figure 2). In contrast to the arcillite-grown lettuce, hydroponically grown plants were significantly affected by watering with all three concentrations (31 ppb, 125 ppb, and 500 ppb) of silver-treated water as assessed by the growth metrics of plant height (at *p* ≤ 0.001 for all), plant area (at *p* ≤ 0.001 for all), shoot fresh mass (at *p* ≤ 0.001 for all), leaf area (at *p* ≤ 0.001 for all), leaf number (at *p* ≤ 0.001 for all), and shoot dry mass (at *p* ≤ 0.001 for all). There was no significant difference in chlorophyll content for any of the silver-treated hydroponic-grown plants as compared to the control, yet the silver treatment concentrations of 125 ppb and 500 ppb showed a significant increase (at *p* ≤ 0.001) in anthocyanin content—a common plant stress response (Figure 3).

### 3.2. Silver and Elemental Nutrient Content of Edible Biomass

The arcillite-grown plants watered with the three concentrations of silver-treated water showed no uptake of silver into the edible biomass of the leaves or any effect on other elements such as calcium, potassium, iron, phosphorus, or sulfur (Figure 4). In contrast, the hydroponic plants grown under the 125 ppb (at *p* ≤ 0.05) and 500 ppb (not significant) concentrations of silver-treated water showed uptake of silver into the edible biomass of plants. This uptake of silver saw an inverse relationship with the ability to uptake calcium (at *p* ≤ 0.05 for 125 ppb and *p* ≤ 0.01 for 500 ppb), potassium (at *p* ≤ 0.001 for both), phosphorus (at *p* ≤ 0.01 for both), and sulfur (at *p* ≤ 0.01 for both) in the hydroponic 125 ppb and 500 ppb silver treatment conditions as compared to the control (0 ppb silver), thus affecting other available nutrients in the plant (Figure 5). The following vitamins were not detectable in either treatment or growth setup: thiamine HCl, cis-menaquinone 7 (cis-MK7, vitamin K2), menaquinone 4 (MK4, vitamin K2), and trans-menaquinone 7 (trans-MK7, vitamin K2). Vitamin K1 (phylloquinone) and ascorbic acid (vitamin C) were detectable but showed no significant difference across treatments according to a *t*-test.

### 3.3. Microbial Diversity Assessed by Shotgun Metagenomic

#### 3.3.1. Alpha Diversity

Whole shotgun metagenomics data were obtained for the microbial consortia of 312 samples consisting of shoots, roots, and irrigation water from plants grown in arcillite and hydroponics system watered with 0 ppb, 31 ppb, 125 ppb, or 500 ppb silver-treated water (Figure 6). Of these 312 samples, 126 produced greater than one microbial genus within the sample (Appendix A). A total of 43 of the 48 arcillite root samples produced greater than one classified genus; whereas only 6 of the 48 arcillite shoot samples produced greater than one classified genus. A total of 10 out of 12 of the arcillite water controls passed this threshold. A total of 27 of the 48 hydroponic root samples and 5 of the 48 hydroponic shoot samples produced greater than one classified genus. A total of 36 of the 108 hydroponic water tray samples collected over four weeks passed this threshold. For the arcillite samples, the data illustrate an overall trend in the Shannon index where the alpha diversity, or the diversity of microbes within the sample, was similar across control and treatments (Figure 6A–C). The alpha diversity was the highest in the root samples from arcillite experiments (Figure 6A) when compared to arcillite-grown shoot samples (Figure 6C) and hydroponic-grown root (Figure 6D) and shoot samples. Shoots from both arcillite (Figure 6C) and hydroponic-grown lettuce were equally low in alpha diversity. For the hydroponic root (Figure 6D) and tray water samples at 28 days (Figure 6E), the data illustrated an overall trend under the Shannon index where the alpha diversity increased as the silver dosage increased (Figure 6). The statistical comparison between the 31 ppb treatment to 500 ppb treatment showed a *p*-value of less than 0.05; however, the *p*-values between each of the treatments to control were not statistically significant for the hydroponic root samples (Figure 6D). Conversely, the statistical comparison between each of the treatments to control treatments for the hydroponics water trays each showed a *p*-value of less than 0.05 with no significant difference between treatments (Figure 6E). Weekly sampling of the hydroponic water tray illustrates that there is no significant difference in alpha diversity between the control and week 4 when the silver treatments are combined for the respective weeks. Week 1, 2, and 3 silver treatments are each statistically lower in alpha diversity than the 0 ppb silver-treated control. This illustrates that the alpha diversity of the hydroponic water tray increases over time irrespective of silver concentration (Figure 6F) and further suggests the microbial community is being shaped by and becoming more resilient to silver treatment over time.

#### 3.3.2. Relative Abundance of Microbes

In arcillite samples, the microbial genera differ in amount by sample type, whether the source is the irrigation water, root, or shoot, but not by silver treatment concentration. Irrigation water samples have either *Shingomonas* or *Methylobacterium*. However, the genera that predominate in the water samples are not the same as those that predominate in the arcillite-grown lettuce shoot or root samples. This leads to the conclusion that the source of the microbes in the arcillite-grown root and shoot samples is likely seed-derived versus derived from microbes in the silver irrigation water. The diversity of the arcillite-grown shoot samples is low and characterized by a higher abundance of the genera *Pseudomonas* and *Maxilla*, which mirrors the genera that are in the highest relative abundance in the arcillite-grown root samples. Finally, the diversity of the root samples is greatest harboring the genera *Pseudomonas* and *Maxilla*, in addition to *Ralstonia*, *Methylobacterium*, *Cupriavidus*, and *Herbasprillium*, among others (Figure 7).

For the microbial genera present in the hydroponic-grown root samples, both the 125 ppb and 500 ppb ionic silver treatment displayed representatives of *Shingomonas*, *Sphingobium*, *Ralstonia*, *Novoshingobium*, and *Cupriavidus* (Figure 7). These genera are also present in the 125 ppb and 500 ppb silver-treated water from the hydroponic water tray but in different relative abundances as compared to the corresponding hydroponic root samples at the same silver concentration. For example, representatives of *Cupriavidus* increase in the 500 ppb silver hydroponic root samples as compared to the relative abundance of the genus in the 500 ppb silver hydroponics tray water sample, whereas representatives of *Novoshingobium* appear to decrease in the 500 ppb silver hydroponics roots samples when compared to the 500 ppb silver hydroponic tray water sample. This suggests that the source of microbes in the roots and shoots of the hydroponic setup is largely driven by the microbes present in the irrigation water of the tray.

### 3.4. Metagenomic Pathway Enrichment

The microbial genetic pathways for each of the 126 metagenomic samples were summarized by MetaPhlAn3. The relative pathway abundances for each of the silver treatments (31 ppb, 125 ppb, 500 ppb silver) were compared to the relative pathway abundances of the control (0 ppb silver) samples within the arcillite and hydroponics setups. Each treatment type has a replicate of at least three samples (Appendix A). The differentially expressed pathways in the silver treatments of the arcillite setup illustrate pathway commonalities (Appendix A). All arcillite-grown silver treatment samples show a greater than 2.5 log_2_FC for ECASYN-PWY: enterobacterial common antigen biosynthesis. In addition, the bacteria represented in the 31 ppb and 125 ppb treatments share an enrichment in pathways PWY-8173, PWY-8174, and PWY-8175 associated with branched-chain fatty-acid biosynthesis. The pathway common to the 125 ppb and 500 ppb treatments includes a decrease in NAD-BIOSYNTHESIS-II: NAD salvage pathway III (to nicotinamide riboside), commonly affected by carbon and nitrogen availability. Furthermore, the 31 ppb and 500 ppb treatments both hold in common an enrichment of the PWY-5896 superpathway of menaquinol-10 biosynthesis involved in the generation of vitamin K2.

In the 31 ppb hydroponic-grown root samples, no bacterial genomic pathways were enriched as compared to the control hydroponic root sample (Figure 8A). However, thirteen differentially enriched pathways for the 125 ppb silver-treated hydroponic root samples overlapped with the differentially enriched pathways of the 500 ppb silver-treated hydroponic root samples when compared to the control hydroponic root samples (Figure 8A). These enriched pathways are present in the genomes of microbes, which increased in both diversity and abundance within the plant roots as the silver concentration increased in the hydroponic tray water. Such enriched pathways include BRANCHED-CHAIN-AA-SYN-PWY: branched-chain amino acid biosynthesis encompassing valine (VALSYN-PWY) and isoleucine (ILEUSYN-PWY) biosynthesis as well as fatty-acid biosynthesis (PWY-7664: oleate biosynthesis IV (anaerobic), PWY0-862: (5Z)-dodecenoate biosynthesis I, and PWY-6282: palmitoleate biosynthesis I (from (5Z)-dodec-5-enoate)) and elongation. This is accompanied by enrichment in the ability to utilize the glyoxylate cycle (GLYOXYLATE-BYPASS: glyoxylate cycle) and access substrates such as acetyl-CoA, fatty acids, alcohols, esters, waxes, alkenes, and methylated compounds. PWY66-389: phytol degradation is also enriched, likely allowing for this plant-derived compound to be metabolized by the microbe (Figure 8B). The 12 pathways specific to the 500 ppb silver treatment indicate mechanisms to support alternative metabolic strategies and stress resistance. Interestingly, the PPGPPMET-PWY: ppGpp metabolism pathway is also enriched in the 500 ppb silver treatment. ppGpp (guanosine 3′-diphosphate 5′-diphosphate) is associated with a stringent response global regulatory system that operates under conditions of nutrient or energy starvation or other environmental stresses (Figure 8C). These pathways suggest possible mechanisms that facilitate microbes as they exhibit an increased association with plant roots, possibly as biofilms, upon increasing silver concentration in the hydroponic tray.

## 4. Discussion

Food-related silver studies have largely investigated the effect of AgNPs rather than ionic silver. Space exploration water systems may use immobilized AgClNPs to deliver ionic silver (Ag^+^) to spacecraft potable water at a dosage between 200 to 500 ppb (0.2–0.5 mg/L). This specified amount is due to the effective dose for microbial biocide activity as well as the need to maintain a high enough silver concentration given that silver can precipitate out of solution. Therefore, the final concentration of silver that will be dispensed from the potable water dispenser is currently unknown and may vary. The recommendation from the Environmental Protection Agency (EPA) is for the concentration of silver in drinking water to not exceed 0.10 milligrams per liter of water (100 ppb) because of the skin discoloration that may occur [25]. Here, a range of residual silver concentrations of 0 ppb, 31 ppb, 125 ppb, and 500 ppb were tested to encompass the highest potential dose in an exploration water system. Here it was investigated how Ag^+^ biocide irrigation water affected plants grown in both a substrate-based arcillite setup and a substrate-less hydroponics setup. Being a clay material, arcillite has a high cation-exchange capacity for silver capture. In a hydroponic setup with very little cation ion exchange surface, there is little opportunity for the positively charged silver ion to precipitate out of the solution onto a substrate, such as in the negatively charged arcillite clay particles. Therefore, we hypothesized that the hydroponically grown plants would be more affected by silver content due to the direct interaction of the seed and root surfaces with the silver in the water.

Visual observation showed that lettuce grown in arcillite under all three silver water treatments germinated at an equal rate and supported biomass growth on a scale equal to that of the control. The growth metrics of plant height, plant area, shoot fresh mass, leaf area, leaf number, and shoot dry mass supported this observation with none of the metrics from the silver-treated plants significantly different from the control plants. However, lettuce grown in a hydroponic setup with silver showed slower germination and severely stunted growth under 125 ppb and 500 ppb treatments and with variable growth defects at 31 ppb. The same growth metrics as above support this observation. A previous study evaluating lettuce exposure to AgNO_3_ observed stunted growth at as low as 0.01 mg/L (10 ppb) in a hydroponics system [12]. In this study, anthocyanin content increased in the 125 ppb and 500 ppb silver treatments and this translated visually to a high red pigmentation in the smaller plants of the hydroponics experiments. Prior work in Arabidopsis has also observed this dose-dependent increase in anthocyanin accumulation as an antioxidant response under silver stress [33].

Given the significant effect the silver treatments had on the growth metrics of the hydroponic-grown lettuce plants, the uptake of silver by the lettuce plants was investigated next. No silver was detected in the leaves of lettuce grown in the substrate-based arcillite system; however, silver was detected in the substrate-less hydroponic setup in the 31 ppb, 125 ppb, and 500 ppb silver water treatments. These amounts resulted in 7 μg/g dry mass (7000 ppb) under 125 ppb treatment and 4 μg/g dry mass (4000 ppb) silver under 500 ppb treatment. In food items, the European Chemicals Agency (ECHA) has established an acceptable daily intake (ADI) of 0.9 μg silver ions/kg (0.9 ppb) of body weight per day [34]. According to this guideline and given that 0.2 g of shoot dry biomass (or 5 g of shoot fresh biomass) yields an average of 1.1 μg of silver, a 50 kg individual could consume up to ~40 of the silver stunted plants without exceeding this daily limit. Yet these plants were so severely stunted that it is impractical to recommend growing them hydroponically with such levels of silver in the water. Elemental analysis of the control and silver-treated plants in the hydroponic setup showed an inverse relationship between silver uptake and the uptake of calcium, potassium, iron, phosphorus, and sulfur in the lettuce leaves, as previously observed in radish [14]. The stunted growth of the lettuce plants demonstrates how toxic silver can be in an unbuffered hydroponic system.

The effect on the lettuce microbiome was also assessed for both the arcillite and hydroponic systems in response to increasing concentrations of silver in the irrigation water. It was observed that the alpha diversity, or diversity within each sample, was similar for the root control and silver-treated root samples in the substrate-based arcillite setup. The root samples from the arcillite setup had an overall higher alpha diversity than the roots from the hydroponic setup. The shoots from the substrate-based arcillite setup and the substrate-less hydroponics setup had the lowest diversity when compared to the diversity of the microbes associated with roots and the irrigation water from both setups. Furthermore, the hydroponic and arcillite shoot samples reflected the microbial genera that were most abundant in root samples from the respective silver treatments.

The irrigation water for the substrate-less hydroponics setup had a higher diversity in the control samples than in the three silver water treatments. Among the three biocide concentrations, the alpha diversity is highest at the 500 ppb silver concentration. Taking a closer look at the roots treated with the three silver water concentrations of 31 ppb, 125 ppb, and 500 ppb in the hydroponics setup, the alpha diversity is lowest in the control and progressively increases as the silver concentration increases. Similarly, the alpha diversity increases in the hydroponic water trays over time for all silver treatments. This trend persisted despite weekly water changeouts of the water trays done to deliver fresh biocide and nutrients. The trend signifies that silver has a microbial biocide effect in the hydroponic water tray but is also selecting for potentially resistant microbes over time.

Little is currently known regarding the microbial diversity and response to ionic silver exposure in hydroponic systems. The observed increase in microbial diversity for the plant roots treated with increasing silver concentrations stood out in this dataset because the existing literature on heavy metal contamination in the soil environment reports a decrease in microbial diversity of the root zone microbiome upon exposure [35]. More studies are necessary to understand the intersecting trends between environmentally distinct soil-based outdoor agriculture and indoor controlled environment agriculture. However, research focused on the plant rhizosphere in soils reports on what is called the rhizosphere effect defined as biological, chemical, and physical changes that occur due to root exudates and rhizodeposition (sloughing of the root cap) [36]. Plant roots in this study are likely exhibiting a rhizosphere effect on the hydroponics system by attracting microbes with plant exudates released in response to high silver exposure. The diversity and abundance trends of this study are summarized in Figure 9.

The roots in the 125 ppb and 500 ppb silver treatments of the hydroponic setup were predominated by the genera *Shingomonas*, *Sphingobium*, *Ralstonia*, *Novoshingobium*, and *Cupriavidus.* Each of these genera was also found in the hydroponics irrigation water at the time of harvest but at differing relative abundances. The literature on soil-based agriculture similarly reports the observation of heavy metal exposure and the selection and subsequent proliferation of metal-tolerant species [37]. The genera selected in heavy metal soils include representatives of *Pseudomonas* and *Acinetobacter* [38] which are distinct from the predominately water-borne genera that are enriched in our study. Specifically, we observed a selective pressure for *Cupriavidus* from the hydroponic water tray to associate with the plant root. The *Cupriavidus* genus is noted for its ability to grow in the presence of heavy metals [39]. This observation has interesting implications for the accumulation of resistant species if silver is used as the sole biocide in the water system.

Finally, the enriched bacterial genomic pathways in the arcillite and hydroponic-grown lettuce for each silver irrigation treatment as compared to the respective 0 ppb silver control in each setup were examined. Here it was found that each of the three silver treatments in the arcillite growth setup showed enrichment in the ECASYN-PWY: enterobacterial common antigen (ECA) biosynthesis pathways in the lettuce roots as compared to the control with 0 ppb silver. The ECA is an outer membrane glycolipid shared by all members of the *Enterobacteriaceae* family naturally found in the lettuce microbiome [40]—a family that is not represented by the top ten genera across all samples. In contrast, in the hydroponics silver-treated lettuce roots, there is complete overlap in the enriched pathways for the 125 ppb treatment with the 500 ppb enriched pathways. In both these conditions, a stunted growth phenotype was observed. The enriched pathways illustrate that fatty-acid biosynthesis and the ability to shift metabolism may help microbes under high silver treatments to associate with plant roots to overcome silver stress and access nutrients. Although the exact mechanism of action for silver as an antibacterial is not completely understood, prior work indicates that silver particles cause damage to the bacterial cell membrane and this in turn affects intracellular metabolic activity [41,42,43]. Furthermore, Chen et al. 2019 showed that the introduction of Ag^+^ produced pressure on antibiotic resistance genes similar to that of an antibiotic [44]. This present study did not show an enrichment in any antibiotic resistance genes, however, a microbial community of constituents with an enhanced ability to generate fatty acids may be able to counter damage to the cell membrane caused by silver. In addition, an enhancement in a diversity of metabolic strategies may help microbes to shelter on plant roots as biofilms that feed off plant exudates. Interestingly, in the 500 ppb silver treatment the ppGpp pathway—a bacterial stress response alarmone [45] and transcriptional regulator of the plastid-encoded RNA polymerase in plant chloroplasts [46]—was enriched. Future work could investigate the metatranscriptomes of the lettuce root-associated microbiome to confirm the role of these pathways in response to silver. Furthermore, the plant host transcriptome and metabolome can be collected in parallel to correlate microbiome activity to plant response.

## 5. Conclusions

Spacecraft may utilize controlled-release silver technology in potable water systems. Imparted silver ions will not be removed from the potable water effluent. Until this study, there was no existing evaluation of ionic silver levels on the space crop growth, plant nutrient content, and root zone microbiome. As a result, there were no established guidelines for its use in the space crop production setting. Currently, plant chambers such as the Advanced Plant Habitat on the International Space Station recycle their own transpired water but still require potable water to prime their water tanks and plumbing, and for occasional water volume replenishment. From this work, we determined that in a substrate-based system such as arcillite, lettuce plant growth, nutrient content, microbial diversity, and abundance are not affected by irrigation water with as high as 500 ppb ionic silver. Lettuce plants exposed to silver-treated water showed reduced growth for shoots at 31 ppb silver and severely stunted growth at 125 and 500 ppb when grown in a hydroponics system. Leaves from the hydroponic system showed the accumulation of silver and a reduced uptake of other essential elements as the silver concentration increased. It was observed that the number of microbes associated with roots and water from the growth trays of the hydroponic system increased as the concentration of silver increased. The types of microbes found on the roots and in the trays were similar but present in different relative amounts. This suggests that ionic silver exerted selective pressure on the irrigation water and the lettuce root microbiome in the hydroponics system. Future space crop production systems that lack cation-exchange capacity will need to remove silver to below 31 ppb to ensure plant health and adequate biomass for crop production. However, this biocide effect on water systems by silver does not preclude microbial resistance from occurring. In fact, subinhibitory amounts of silver over time can lead to greater microbial resistance to silver and the buildup of microbial biofilms, and this is an area for further investigation.

## Figures and Tables

**Figure 1 microorganisms-12-00515-f001:**
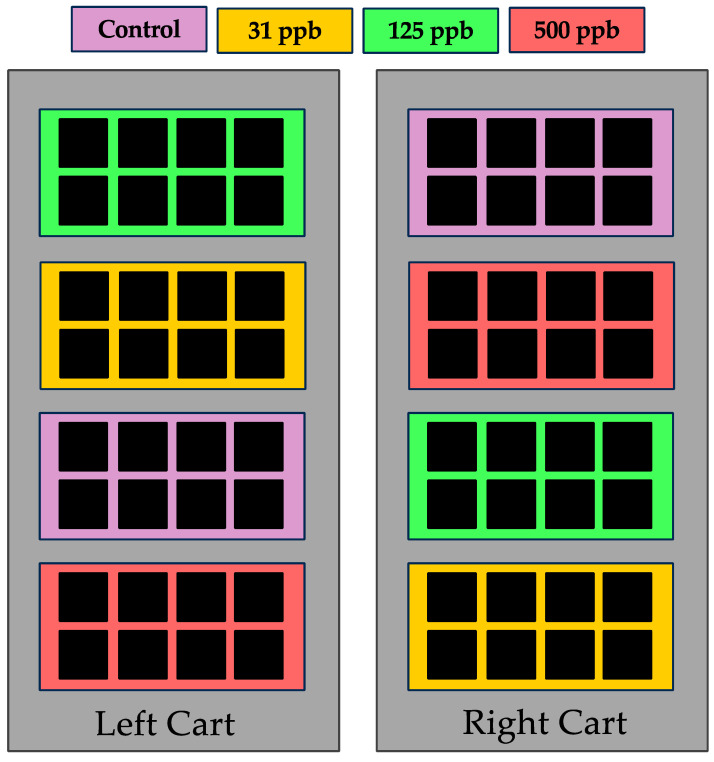
Experimental design sample layout for one replicate of three grow-outs per experiment.

**Figure 2 microorganisms-12-00515-f002:**
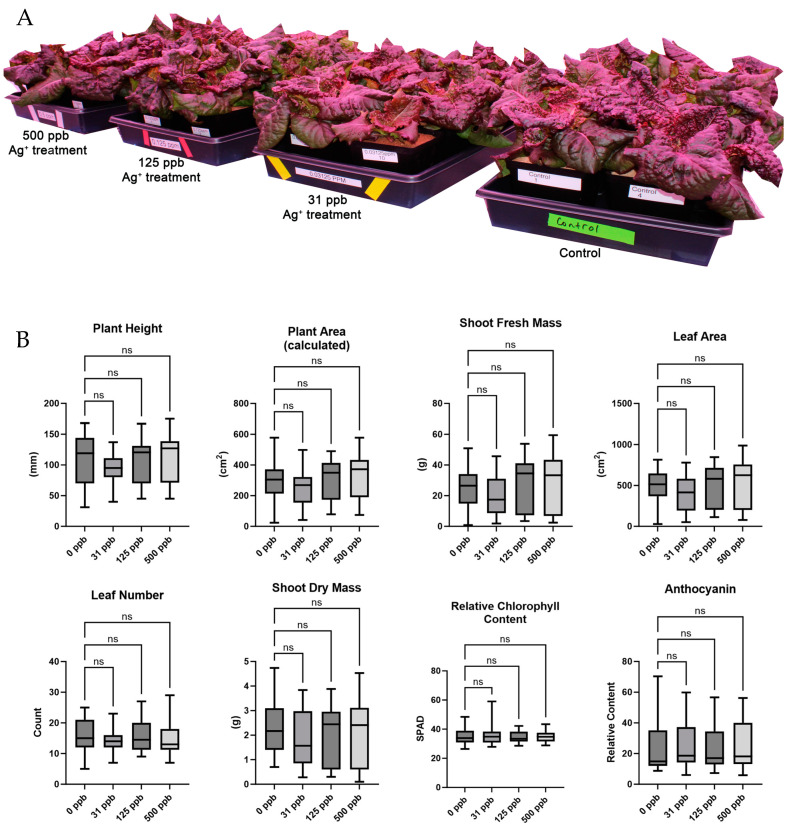
(**A**) Image of arcillite-grown plants at 28 DAP. (**B**) Harvest metrics for arcillite grown lettuce averaged for each of the three treatment grow outs. ns is not significant.

**Figure 3 microorganisms-12-00515-f003:**
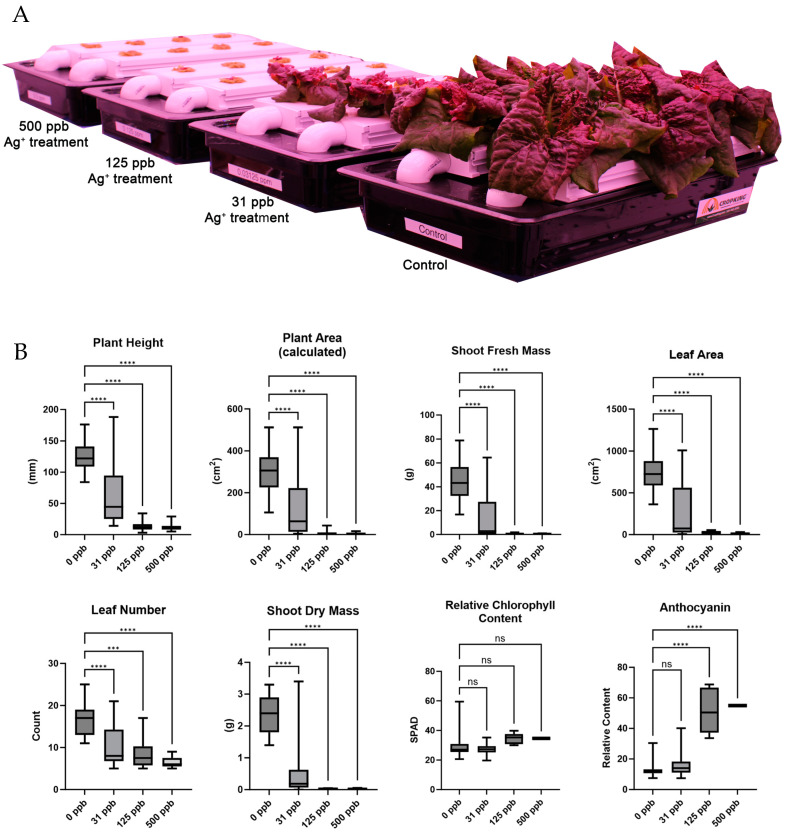
(**A**) Image of hydroponic-grown plants at 28 DAP. (**B**) Harvest metrics for hydroponic-grown lettuce results averaged for each of the three treatment grow outs. **** is *p* ≤ 0.0001, *** is *p* ≤ 0.001, ns is not significant.

**Figure 4 microorganisms-12-00515-f004:**
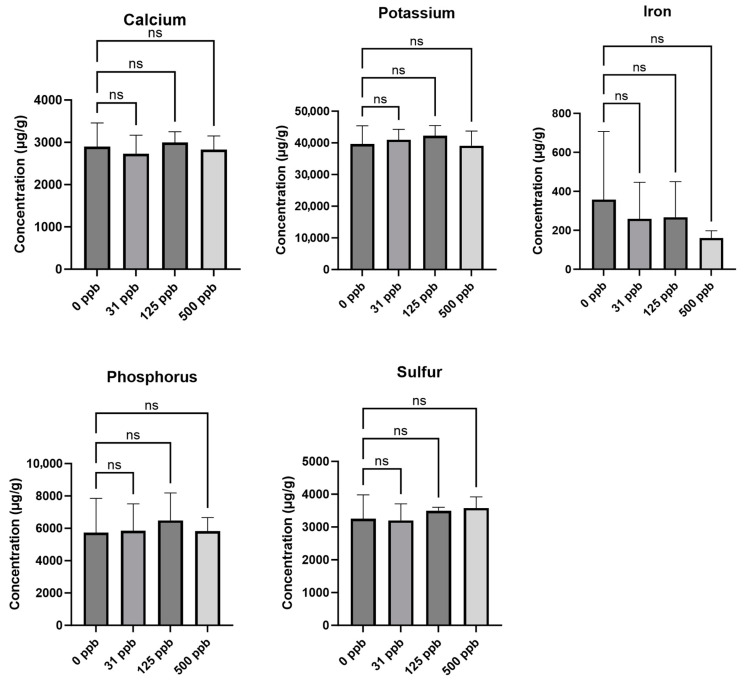
Lettuce leaf elemental content of arcillite-grown plants at 28 DAP average for each of the three treatment grow-outs. No silver was detected. ns is not significant.

**Figure 5 microorganisms-12-00515-f005:**
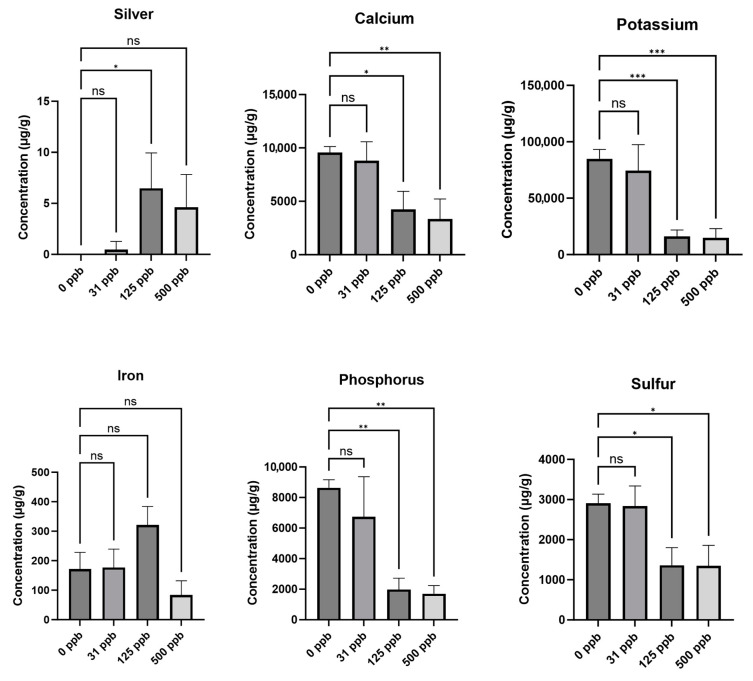
Lettuce leaf silver and elemental content of hydroponically grown plants at 28 DAP averaged for each of the three treatment grow-outs. *** is *p* ≤ 0.001; ** is *p* ≤ 0.01; * is *p* ≤ 0.05; ns is not significant.

**Figure 6 microorganisms-12-00515-f006:**
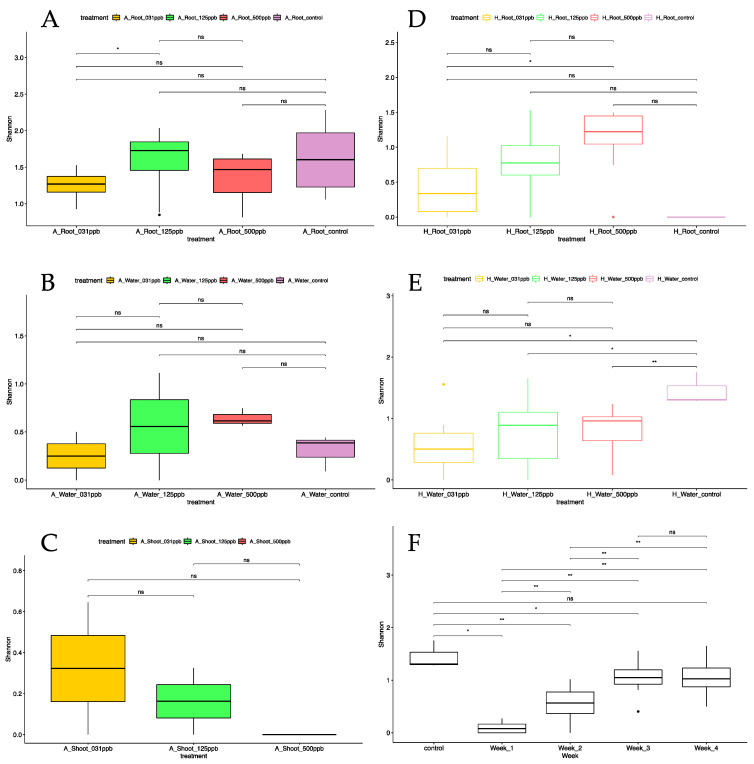
Boxplots of Shannon index alpha diversity metrics for silver-treated: (**A**) arcillite roots; (**B**) arcillite water; (**C**) arcillite shoots; (**D**) hydroponic roots; (**E**) hydroponic water, each at 28 DAP; and (**F**) weekly hydroponic water sampling averaged for all replicates over the three grow-outs pairwise *p*-values. ** is *p* ≤ 0.01; * is *p* ≤ 0.05; ns is not significant.

**Figure 7 microorganisms-12-00515-f007:**
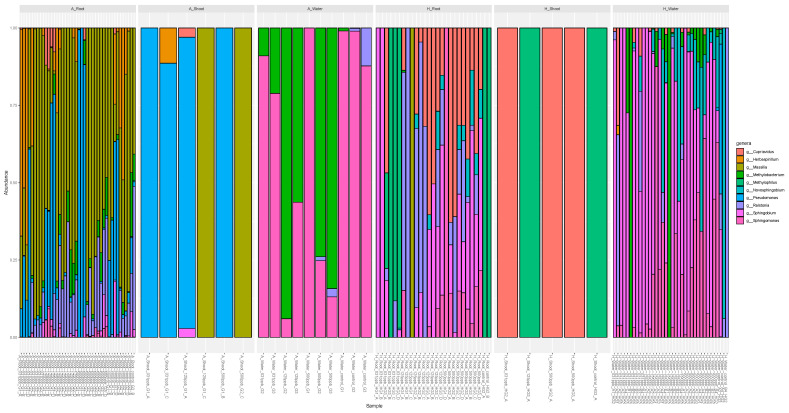
Relative abundances of the top 10 microbes present across silver-treated arcillite, hydroponics (roots and shoots), and water samples. Sample labeling key for A_Root, A_Shoot, A_Water, H_Root, H_Shoot: [arcillite (A) or hydroponic (H)]_[root, shoot, or water]_[ppb of silver]_[experimental grow out (G or HG): 1,2, or 3]_[ experimental sample: A,B,C or D.]. Sample labeling key for H_Water: [hydroponic (H)]_[ppb of silver]_[experimental sample: A,B,C or D with water sample week number 1,2,3 or 4}_[experimental grow out (HG): 1,2, or 3].

**Figure 8 microorganisms-12-00515-f008:**
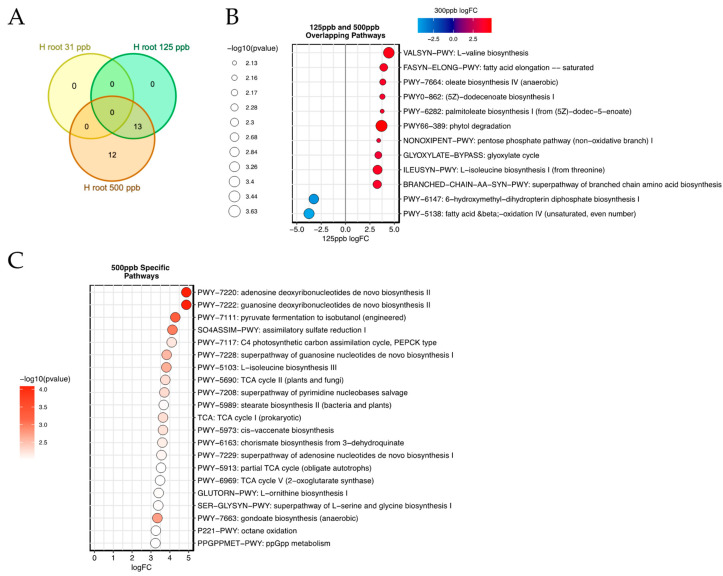
(**A**) Venn diagram illustrating the significantly enriched pathways for the silver-treated roots of the lettuce hydroponics samples as compared to the 0 ppb silver root control and the overlap of the 3 treatment comparisons. (**B**) The log_2_FC of the thirteen enriched pathways specific to the 125 ppb silver-treated roots are plotted on the x-axis and the −log_10_ (*p*-values) of the 125 ppb silver-treated roots are plotted on the y-axis. Also represented on this plot are the log_2_FC values of the thirteen overlapping pathways also enriched for the 500 ppb silver-treated roots. This is represented as a shade between red or blue and the −log10 (*p*-values) represented by the size of the circle. (**C**) The log_2_FC of the twelve enriched pathways specific to the 500 ppb silver-treated roots are plotted on the x-axis and the −log_10_ (*p*-values) of the 500 ppb silver-treated roots are plotted on the y-axis.

**Figure 9 microorganisms-12-00515-f009:**
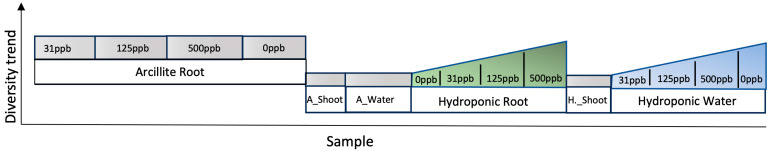
Summary of diversity and abundance trends for the microbiome of lettuce roots and shoots and irrigation water from arcillite and hydroponic setups treated with 31 ppb,125 ppb, or 500 ppb silver water.

## Data Availability

Whole genome metagenomics data are available at NASA’s Genelab.

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
