# Peer review of "Substrate Matters: Ionic Silver Alters Lettuce Growth, Nutrient Uptake, and Root Microbiome in a Hydroponics System"

_microorganisms, 2024, doi:10.3390/microorganisms12030515_

Round 1

Reviewer 1 Report

Comments and Suggestions for Authors

Comments on “Substrate matters: Ionic silver alters lettuce growth, nutrient uptake and root microbiome in a hydroponic system

1.     The introduction should provide more background information on the potential challenges of plant growth in space and the importance of hydroponic systems in space crop production. This will help readers understand the significance of the study.

2.     The objectives of the study should be clearly stated in the introduction or methodology section to guide readers throughout the manuscript.

3.     The methodology section should provide more detailed information on the experimental setup, including why the authors chose the specific concentrations of silver-treated water used and the duration of the experiment.

4.     The results section provides clear findings on the impact of different silver concentrations on lettuce growth and nutrient uptake. However, it would be helpful to include statistical analysis to support the significance of the results.

5.     The discussion section should compare the study findings with previous research on the effects of silver on plant growth and microbial communities. This will help readers understand the novelty and contribution of the study.

6.     The conclusion should summarize the key findings of the study and highlight their implications for future space crop production. Additionally, any limitations of the study should be acknowledged.

7.     Almost all the figures in this manuscript are of low quality and the claims of the authors can not be verified. Kindly improve the resolution of all figures.

8.     The manuscript could be enhanced by providing more information on the specific mechanisms by which silver affects plant growth and root microbiomes. This would add depth to the scientific understanding of the topic.

9.     The language of the manuscript is generally clear and concise, but some sentences could be rephrased for better clarity and readability.

10.  The authors could provide more context on the potential applications of silver-treated water in spaceflight platforms and its advantages over other microbial biocides.

11.  The authors mention the use of cation exchange capacity in hydroponic systems, but it does not explain what this means or why it is relevant. This information should be provided for readers who may not be familiar with the terminology.

12.  Overall, the document provides valuable insights into the impact of ionic silver on lettuce growth and root microbiomes in hydroponic systems. I encourage authors to thoroughly and extensively improve the results and discussion section most especially the microbial part using: 10.1007/s11104-023-06256-4. With some revisions and clarifications, it can be further improved to enhance its scientific soundness and readability.

Comments on the Quality of English Language

Minor language editing required

Author Response

1. The introduction should provide more background information on the potential challenges of plant growth in space and the importance of hydroponic systems in space crop production. This will help readers understand the significance of the study.

Reply: Thank you, this suggestion has been implemented and we believe it has made the introduction much better. We specifically refer you to lines 59 -70 in the revised manuscript.

These lines now read:

"In addition to providing clean water to the crew and other life support system functions, potable water will be used to irrigate future spaceflight crop production systems. Substrate-based watering systems are currently used on the International Space Station (e.g. Veggie and Advanced Plant Habitat (APH)) and use arcillite- a calcinate clay with high cation ion exchange capacity [4,5]. Surface-tension effects exhibited in reduced gravity can lead to thick boundary layers of water around plant roots and in turn lead to root hypoxia [6]. Substrate-free nutrient delivery systems such as hydroponics can reduce waste and up-mass in future missions [7] and can provide efficient absorption of nutrients as roots are directly in contact with the nutrient solution [8]. Recent work to evaluate this efficiency and overcome the challenges of watering plants in the microgravity environment has included spaceflight testing of hydroponics systems such as Plant Water Management (PWM) [9] and eXposed Root On-Orbit Test System (XROOTS) [10]."

2. The objectives of the study should be clearly stated in the introduction or methodology section to guide readers throughout the manuscript.

Reply: Thank you, the objectives of the study are now clearly stated at the end of the introduction in lines 101-114 and recapitulated lines 116-122 of the Materials and Methods. 

Lines 101-114, now read:

"The use of Ag+ may adversely affect plant growth in spaceflight but the choice of a substrate-based or substrate free growth system and silver concentration could mitigate the more extreme effects. One reason for this is due to the high cation exchange capacity of arcillite that allows for adsorption of positively charged ions due to negatively charged surfaces. We expected that arcillite as a substrate with high cation exchange capacity would act like biochar, which holds a lower cation exchange capacity, in its ability to diminish the uptake of toxic metals in plants by immobilizing the metal across the surface of the substrate via ion exchange [23,24]. To that end, it remained to be determined what dosage of Ag+ is acceptable for plant irrigation water under ISS-like environmental conditions (~3000ppm carbon dioxide) and whether Ag+ will need to be removed from potable water before use as space crop irrigation water. To answer this, we asked at what concentration does silver-treated water influence the growth, nutrient content, and microbiome of a space crop grown on an arcillite (calcinated clay) substrate or without a substrate (hydroponically)?"

and lines 116-121, now read:

"In this study, we investigated the effect of Ag+ on lettuce plant growth in International Space Station (ISS)-relevant growth chamber conditions in both the traditional ISS growth medium of arcillite and in a hydroponics set up. The methods of this study were conducted to address the following objective: to determine at what concentration silver-treated water influences the growth, nutrient content, and microbiome of a space crop grown on an arcillite (calcinated clay) substrate or without a substrate (hydroponically)."

3. The methodology section should provide more detailed information on the experimental setup, including why the authors chose the specific concentrations of silver-treated water used and the duration of the experiment.

Reply: The methodology section 2.1 has been updated to provide more information on the experimental setup. The section in lines 123-130 now reads:

"The design specifications for a silver dosing device in a spacecraft potable water systems includes the ability to add between 200 to 500 ppb (0.2-0.5mg/L) of silver. The final amount of silver that will dispense from the potable water dispenser for end use is currently unknown and may vary. Therefore, we selected a dilution series of ionic silver to encompass 1. the highest possible dose (500 ppb) to dispense from the potable water system, 2. a dose (~125 ppb) close to the acceptable drinking water dose limits of silver, which is 100 ppb [25] and 3. a lowest dose (31 ppb) to represent a dose thought to have next to no adverse effects in a substrate-based system as reported in the literature [12]."

And section 2.4 in lines 168-169 now reads:

"Plants were grown for 28 days to achieve a mature specimen, similar to what would be ingested by crew in an ISS operation setting."

4. The results section provides clear findings on the impact of different silver concentrations on lettuce growth and nutrient uptake. However, it would be helpful to include statistical analysis to support the significance of the results.

Reply: Thank you, this was a very good point and the statistical p-values in the figures are now reported in the respective results sections.

5. The discussion section should compare the study findings with previous research on the effects of silver on plant growth and microbial communities. This will help readers understand the novelty and contribution of the study.

Reply: Thank you, this was a very good suggestion and the discussion section has been updated according line for lines 473-485 and lines 486-497

Lines 474-485 now read:

"Little is currently known regarding the microbial diversity and response to ionic silver exposure in hydroponic systems. The observed increase in microbial diversity for the plant roots treated with increasing silver concentrations stood out in this dataset because the existing literature on heavy metal contamination in the soil environment reports a decrease in microbial diversity of the root zone microbiome upon exposure [32].More studies are necessary to understand the intersecting trends between the environmentally distinct soil-based outdoor agriculture and indoor controlled environment agriculture. However, research focused on the plant rhizophere in soils reports on what are called the rhizosphere effects defined as biological, chemical, and physical changes that occur due to root exudates and rhizodeposition (sloughing of the root cap) [33]. Plant roots in this study are likely exhibiting a rhizosphere effect on the hydroponics system by attracting microbes with plant exudates released in response to high silver exposure. Diversity and abundance trends of this study are summarized in Figure 9."

Lines 486-497 now read:

"The roots in the 125 ppb and 500 ppb silver treatments of the hydroponic setup were predominated by the genera Shingomonas, Sphingobium, Ralstonia, Novoshingobium, and Cupriavidus. Each of these genera were also found in the hydroponics irrigation water at the time of harvest but at differing relative abundances. Literature in soil-based agriculture similarly reports the observation of heavy metal exposure and the selection and subsequent proliferation of metal-tolerant species [34] . The genera selected for in heavy metal soils include representatives of Pseudomonas and Acinetobacter [35] that are distinct from the predominately water-borne genera that are in enriched in our study. Specifically, we observed a selective pressure for Cupriavidus from the hydroponic water tray to associate with the plant root. The Cupriavidus genus is noted for the ability to grow in the presence of heavy metals [36]. This observation has interesting implications for the accumulation of resistant species if silver is used as the sole biocide in the water system."

6. The conclusion should summarize the key findings of the study and highlight their implications for future space crop production. Additionally, any limitations of the study should be acknowledged.

Thank you, the conclusion has been updated in lines 534-546. This summary material was once the simple summary that in not required for this issue.

Lines 534-546 now read:

"From this work, we determined that in a substrate-based system such as arcillite, lettuce plant growth, nutrient content and microbial diversity and abundance are not affected by irrigation water with as high as 500 ppb ionic silver. Lettuce plants exposed to silver-treated water showed reduced growth for shoots at 31ppb silver, and severely stunted growth at 125 and 500 ppb when grown in a hydroponics system. Leaves from the hydroponic system showed the accumulation of silver and a reduced uptake of other essential elements as the silver concentration increased. It was observed that the number of microbes associated with roots and water from the growth trays of the hydroponic system increased as the concentration of silver increased. The types of microbes found on the roots and in the trays were similar but present at different relative amounts. This suggests that ionic silver exerted a selective pressure on the irrigation water and the lettuce root microbiome in the hydroponics system. Lettuce plants grown in the arcillite substrate-based system and irrigated with silver-treated water did not show any change in growth, nutrient content or microbiome as compared to the control."

7. Almost all the figures in this manuscript are of low quality and the claims of the authors can not be verified. Kindly improve the resolution of all figures.

Thank you, all figure have been improved and placed in a separate Word document with figure legends.

8. The manuscript could be enhanced by providing more information on the specific mechanisms by which silver affects plant growth and root microbiomes. This would add depth to the scientific understanding of the topic.

Reply: We believe that edits to lines 474-485 may satisfy this request. Additionally, the introductions states how AgNPs affect plant and discusses how as AgNPs approach the small size of ionic silver, the effects become stronger.

9. The language of the manuscript is generally clear and concise, but some sentences could be rephrased for better clarity and readability.

Thank you, the manuscript have been revised and text has been rephrased for better clarity and readability.

10. The authors could provide more context on the potential applications of silver-treated water in spaceflight platforms and its advantages over other microbial biocides.

Reply: In the introduction we have edited the test to highlight the current use of iodine and the advantage of using silver in lines 47-58.

Lines 47-58 now read:

"Ionic silver (Ag+) is being investigated as a residual biocide for use in NASA spacecraft potable water systems on future crewed missions and has been used for many years on the Russian Mir and Russian modules of the International Space Station [1,2]. NASA currently uses iodine (I2) as a biocide in the potable water system on the International Space Station (ISS). Iodine is removed before use because excessive amounts can lead to thyroid dysfunction [3]. The advantage of ionic silver is that it does not need to be removed from potable water before consumption. The design specifications for a silver dosing device in a spacecraft potable water systems includes the ability to add between 200 to 500 ppb (0.2-0.5mg/L) of silver [2]. The final amount of silver that will dispense from the potable water dispenser for end use is currently unknown and may vary due to the propensity for silver to plate out of solutions onto systems materials [3]."

11. The authors mention the use of cation exchange capacity in hydroponic systems, but it does not explain what this means or why it is relevant. This information should be provided for readers who may not be familiar with the terminology.

Thank you for pointing this out and the text has been updated accordingly in lines 101-108, which now read: 

"The use of Ag+may adversely affect plant growth in spaceflight but the choice of a substrate-based or substrate free growth system and silver concentration could mitigate the more extreme effects. One reason for this is due to the high cation exchange capacity of arcillite that allows for adsorption of positively charged ions due to negatively charged surfaces. We expected that arcillite as a substrate with high cation exchange capacity would act like biochar, which holds a lower cation exchange capacity, in its ability to diminish the uptake of toxic metals in plants by immobilizing the metal across the surface of the substrate via ion exchange [23,24]."

12. Overall, the document provides valuable insights into the impact of ionic silver on lettuce growth and root microbiomes in hydroponic systems. I encourage authors to thoroughly and extensively improve the results and discussion section most especially the microbial part using: 10.1007/s11104-023-06256-4. With some revisions and clarifications, it can be further improved to enhance its scientific soundness and readability.

Dear Reviewer, thank you for your thoughtful review of our manuscript. Your suggestions have improved the document. We have improved the microbial part in the discussion and included a citation from the suggested paper in lines 105-107 of the revised manuscript.

Reviewer 2 Report

Comments and Suggestions for Authors

The manuscript presents an interesting topic and an innovative scientific hypothesis. The experiment is well planned and the conclusions are pertinent, clear and concise.

I did not understand how you determined the three concentrations of ionic silver biocide solutions. I think this choice needs to be explained.

The English used is flawless and at a compressible level.

It would have been even more interesting if the data obtained could have been compared with a control in classical culture.

Author Response

1. The manuscript presents an interesting topic and an innovative scientific hypothesis. The experiment is well planned and the conclusions are pertinent, clear and concise.

Reply: Thank you for your approval of the topic and manuscript.

2. I did not understand how you determined the three concentrations of ionic silver biocide solutions. I think this choice needs to be explained.

Reply: Thank you for asking for this clarification. Lines 126-130 now read,

"Therefore, we selected a dilution series of ionic silver to encompass 1. the highest possible dose (500 ppb) to dispense from the potable water system, 2. a dose (~125 ppb) close to the acceptable drinking water dose limits of silver, which is 100 ppb [25] and 3. a lowest dose (31 ppb) to represent a dose thought to have next to no adverse effects in a substrate-based system as reported in the literature [12].

3. The English used is flawless and at a compressible level.

Reply: Thank you, we have however edited for clarity and readability.

4. It would have been even more interesting if the data obtained could have been compared with a control in classical culture.

Reply: This is a good point that we will keep in mind for future experiments comparing arcillite and hydroponics. 

Round 2

Reviewer 1 Report

Comments and Suggestions for Authors

Revised version is better.

Comments on the Quality of English Language

ok